# Novel Biomolecules in the Pathogenesis of Gestational Diabetes Mellitus

**DOI:** 10.3390/ijms222111578

**Published:** 2021-10-27

**Authors:** Monika Ruszała, Magdalena Niebrzydowska, Aleksandra Pilszyk, Żaneta Kimber-Trojnar, Marcin Trojnar, Bożena Leszczyńska-Gorzelak

**Affiliations:** 1Department of Obstetrics and Perinatology, Medical University of Lublin, 20-090 Lublin, Poland; mniebrzydowska7@gmail.com (M.N.); apilszyk@gmail.com (A.P.); b.leszczynska@umlub.pl (B.L.-G.); 2Department of Internal Diseases, Medical University of Lublin, 20-059 Lublin, Poland; marcin.trojnar@umlub.pl

**Keywords:** gestational diabetes mellitus, biomolecules, adipokines, predictor

## Abstract

Gestational diabetes mellitus (GDM) is one of the most common metabolic diseases in pregnant women. Its early diagnosis seems to have a significant impact on the developing fetus, the course of delivery, and the neonatal period. It may also affect the later stages of child development and subsequent complications in the mother. Therefore, the crux of the matter is to find a biopredictor capable of singling out women at risk of developing GDM as early as the very start of pregnancy. Apart from the well-known molecules with a proven and clear-cut role in the pathogenesis of GDM, e.g., adiponectin and leptin, a potential role of newer biomolecules is also emphasized. Less popular and less known factors with different mechanisms of action include: galectins, growth differentiation factor-15, chemerin, omentin-1, osteocalcin, resistin, visfatin, vaspin, irisin, apelin, fatty acid-binding protein 4 (FABP4), fibroblast growth factor 21, and lipocalin-2. The aim of this review is to present the potential and significance of these 13 less known biomolecules in the pathogenesis of GDM. It seems that high levels of FABP4, low levels of irisin, and high levels of under-carboxylated osteocalcin in the serum of pregnant women can be used as predictive markers in the diagnosis of GDM. Hopefully, future clinical trials will be able to determine which biomolecules have the most potential to predict GDM.

## 1. Introduction

Gestational diabetes mellitus (GDM) is one of the most common metabolic diseases encountered in obstetrics, and it is often associated with an increased risk of adverse pregnancy outcomes [1,2]. GDM affects about 9–25% of pregnancies worldwide [3]. The frequency of GDM increases with the age of a pregnant woman. Glucosuria, excessive weight gain, polyhydramnios, and fetal macrosomia during pregnancy are indications for further diagnosis of carbohydrate metabolism disorders [4]. The algorithm for detecting GDM includes performing an oral glucose tolerance test between 24 and 28 weeks of gestation or as soon as possible if the pregnant woman is in a risk group. In clinical practice, a pregnant woman’s fasting blood glucose level is checked at her first visit [5,6]. Patients with a history of overweight or obesity, elevated blood glucose levels or impaired glucose tolerance prior to pregnancy, age over 35 years, family history of diabetes, previously giving birth to a baby with a birth weight larger than 4000 g, obstetrical failures in previous pregnancies (i.e., miscarriages, malformations, intrauterine deaths), with glucose in the urine, or increased amniotic fluid are significantly more likely to develop GDM and diabetes mellitus type 2 (T2DM) in the future [7]. Undiagnosed GDM carries the risks of obstetric complications such as: polyhydramnios, preterm delivery, edema, urinary tract infections, pyelonephritis, fetal macrosomia, or preeclampsia [8]. Thus, GDM therapy should include monitoring of several components, i.e., blood glucose levels, the presence of ketones in urine, weight gain, fetal biometry, and physical activity.

Pathogenesis of GDM is complex and, inter alia, it involves impairment of the action and secretion of insulin. Insulin resistance during pregnancy develops gradually, and finally, in consequence, it leads to hyperglycaemia [9,10]. Glucose, which passes through the placenta, is the main source of energy for the developing fetus. Therefore, the response to the elevated glucose level is increased insulin production by the fetus, which in turn causes muscle tissue hypertrophy, including the heart muscle, fatty tissue, and the liver [11,12]. In consequence, excessive fetal growth (macrosomia) is observed. Moreover, increased levels of hormones in a pregnant woman, i.e., estrogen, progesterone, and placental lactogen, as well as other hormones, increase insulin resistance even further. Surely, excessive body fat, low physical activity, hypertension, age, and family history of diabetes may be conductive to intensify this process [13].

Excessive activation of the adipose tissue plays an important role in the pathogenesis of GDM since it is in the adipose tissue that an inflammatory process is initiated [14,15]. Hyperinsulinemia is a major factor affecting the growth of body fat in pregnant women. Consequently, an excessive body mass accelerates the development of GDM. The adipose tissue is a source of monocytes and macrophages that produce inflammatory factors [16]. The tumor necrosis factor-alpha (TNF-α) has a proapoptotic activity. It affects cell differentiation and cell recruitment. TNF-α also stimulates the synthesis of prostaglandins and initiates oxidative stress [17]. One of the pivotal roles of this cytokine is weakening the expression of the glucose transporter type 4 (GLUT 4) in the adipose tissue, skeletal muscles, and cardiac muscle. It is assumed that more TNF-α is secreted by the placenta than by the adipose tissue [18].

Pregestational obesity and excessive weight gain during pregnancy seem to be the main causes of GDM [15,19]. The adipose tissue is able to secrete hormones, i.e., adipokines, which may contribute to insulin resistance. A “satiety hormone” called leptin impairs insulin-dependent glucose transport to adipocytes. It mediates the increased protein synthesis and may directly impair glucose secretion by the beta cells of the pancreas. During pregnancy, leptin levels increase because leptin is produced by the placenta as well [20,21,22,23,24,25]. The highest increase in leptin concentration is observed in the second trimester of pregnancy, while its concentration decreases rapidly just after delivery. Leptin can act as a metabolic switch connecting the nutritional status of the body to the high energy-consuming processes [26]. Obese pregnant women have significantly elevated plasma leptin concentrations throughout pregnancy compared with non-obese pregnant women throughout pregnancy. The high plasma leptin levels may be potentiated by leptin resistance at the central level [27]. The effects of placental leptin on the mother may contribute to the endocrine-mediated alterations in the energy balance, such as mobilization of maternal fat, which could further aggravate the insulin resistance associated with pregnancy and the onset of GDM [28].

The placenta is the organ linking the mother to the fetus [29]. As an endocrine organ, it secretes many cytokines and adipokines, including adiponectin, which mediate fetal development and maternal metabolism [18,30]. In the presence of placental hormones, the body cells become insensitive to insulin, which causes an increase in blood glucose levels. As the baby grows in size, the placenta produces more and more hormones, which results in hyperglycaemia.

In addition to the well-known molecules with a proven important role in the pathogenesis of GDM, including adiponectin and leptin [18,19,22,23,24,25,30], the potential role of certain novel biomolecules is also emphasized. The less popular and less known biomolecules of interest include: galectins, GDF-15, chemerin, omentin-1, osteocalcin, resistin, visfatin, vaspin, irisin, apelin, FABP4, FGF21, and lipocalin-2. Therefore, the aim of this review is to present these 13 novel biomolecules and their potential significance in GDM.

## 2. Biomolecules

### 2.1. Galectins

Galectins are a large family belonging to the lectins with relatively broad specificity [31]. They have diverse functions, including mediating cell-cell interactions, cell-matrix adhesion, and trans-membrane signaling. During pregnancy, galectins participate in maternal immune responses, angiogenesis, and placentation [32]. A total of 16 classes of galectines are distinguished, of which 10 are found in humans. Most of them are located in the placental tissue. Some galectines such as galectin-1 and -3 were described to be significantly decreased in the placentas of GDM patients. The role of galectin 2 is still uncertain [33].

Some studies indicate the potential involvement of galectin 3 in the pathogenesis of GDM. It is assumed that this protein that belongs to a family of β-galactoside binding proteins expresses pro-fibrotic, proinflammatory features and promotes angiogenesis through new capillary formation [34]. Galectin 3 is also known as Mac-2, CBP-35, RL29, or eBP. Its expression has been demonstrated in many cell types, including neutrophils, macrophages, mast cells, fibroblasts, and osteoclasts. It is modulated by estrogens and progesterone [35]. Increased levels of galectin 3 favor the development of insulin resistance. There is a positive correlation between galectin 3 level, fasting glucose, and fasting insulin [36]. In addition, a significant positive correlation was found between galectin 3, gestational age, and current BMI. Due to this fact, it may be used in the first trimester as a potential factor to predict the development of diabetes [37].

The second representative of the lectin family, Galectin 2, is highly concentrated in both the fetal syncytiotrophoblast and maternal decidua [38]. Additionally, there is also expression in extra villous trophoblast cells and fetal endothelial cells. The main role of galectin 2 is immunomodulation via anti-inflammatory actions [39]. Furthermore, this biomolecule is also involved in direct insulin secretion of pancreatic beta cells. Its concentration increases with elevated fasting glucose and fasting insulin levels. Galectin 2 is suspected of having an impact on macrophages M1 and M2 by its polarization and releasing interleukin 6 (IL-6) and TNF-α, which generates insulin resistance [40].

Galectin 1 plays a role in creating immune tolerance in pregnancy (expressing T cells CD3+, CD4+, CD8+, NK cells CD56+), angiogenesis, syncytium formation, and significantly increases during implantation. A decreased level of galectin 1 during pregnancy is often associated with adverse pregnancy outcomes such as abortion or PE [38]. Some studies describe that reduced peripheral levels of galectin 1 would affect the cytokine imbalance, ensuring the Th2 predominance necessary to maintain pregnancy in GDM mothers [41].

Despite some researches available on the role of galectins in the pathogenesis of diabetes, additional studies are needed. Metabolic and hormone alterations should be exhaustively analyzed to assess the possibility of using galectins as a predictor of the occurrence of GDM in early pregnancy to prevent patients from future complications.

### 2.2. GDF-15

GDF-15, which belongs to transforming growth factor beta (TGF-beta) biomolecules and was first described as macrophage inhibitory cytokine-1, placental bone morphogenic protein (PLAB), placental transforming growth factor beta (PTGFB), or NSAID-activated gene-1 (NAG-1), seems to take part in inflammation, cell repair, and their growth after tissue injury [42]. This biomolecule is often significantly elevated in septic patients with bacterial or viral infections or with cardiovascular diseases [43]. GDF-15 is found in the placenta and the fetal membrane [44]. It can also be secreted in the bladder, prostate, stomach, and duodenum [45]. Its highest concentration is detected at the beginning of the third trimester, especially in patients with GDM [46]. GDF-15 secretion is stimulated by high blood glucose levels as a protective factor from glucose intolerance [47]. Some studies highlight the relationship between the risk of GDM and increased GDF-15 levels. In the literature, there is no relationship between GDF-15 levels and adverse perinatal outcomes. GDF-15 levels are influenced by maternal age, BMI, smoking, environmental factors, and gestational age [46]. In addition, there is also a positive correlation between GDF-15 and fasting blood glucose (FBG), 1 h postprandial glucose (1 h-PG), 2 h postprandial glucose (2 h-PG) hemoglobin A1C (HbA1c), and area under curve of glucose (AUCG) in OGTT [47]. The mechanism of GDF-15 in patients with glucose metabolism disorders is still unclear. P53 is probably responsible for GDF-15 expression [48]. Petersen et al. revealed that GDF-15 is likely a myomitokine because it originates from mitochondria in skeletal muscle [49]. In support of this assumption, it was noted that elderly patients developed insulin resistance due to decreased mitochondrial oxidative phosphorylation. Insulin resistance is associated with a decrease in M2 macrophages [50]. GDF-15 promotes the polarization of macrophages to the M2 phenotype in adipose and prevents insulin resistance [51]. It is often described that GDF-15 may improve lipid metabolism through thermogenesis, lipolysis, and oxidation [52]. In addition, in some studies, GDF-15 is described as an appetite suppressor. In consequence, lipid accumulation in the body may decrease.

The role of GDF-15 is still unknown. Because of all these properties, GDF-15 could be a predictor for the future development of diabetes mellitus in pregnant women. Future studies need to be performed to examine whether GDF-15 is released due to the inflammatory process or as a reason associated with carbohydrate imbalance.

### 2.3. Chemerin

Chemerin belongs to a group of novel adipokines believed to have an important role in the regulation of glucose and lipid metabolism. It is a product of the retinoic acid receptor responder 2 (RARRES2) gene located on chromosome 7 [53]. Initially, it is synthesized as an inactive preproprotein containing 163 amino acids. Thereafter, a hydrophobic messenger peptide, consisting of 20 amino acids, is detached, while the produced proprotein is released from the cell. Proteases are responsible for the activation and formation of the final form of chemerin [54]

Chemerin mRNA has been detected in many animal species, including humans, rodents, bovine, and poultry [55]. In humans, it is produced mainly by white adipose tissue (visceral and subcutaneous), liver, and placenta and to a lesser extent by brown adipose tissue, lungs, skeletal muscles, kidneys, ovaries, heart, and adrenal glands [54,55,56,57]. Its physiological functions involve regulation of blood pressure, immune functions, angiogenesis, and inflammation [58]. Moreover, it has endocrine, paracrine, and autocrine effects [54].

It seems that its increased secretion is related to many metabolic and cardiovascular pathologies and contributes to the development of inflammation [59,60]. Chemerin levels are correlated with BMI, obesity, lipid serum levels, and blood pressure [56,61]. According to some studies, increased chemerin secretion occurs in T2DM and insulin resistance, as reflected by its correlation with the HOMA-IR [62]. Insulin resistance is a condition characterized by decreased insulin sensitivity in peripheral tissues. It is believed that one of its promoters is the inflammatory process, which in turn can be stimulated by chemerin. Its receptors appear to be associated with the recruitment of macrophages, dendritic cells, and activation of neutrophils [53,63]. Chemerin is also involved in insulin metabolism. Insulin significantly enhances chemerin secretion from adipose tissue, and chemerin affects insulin transmission and glucose degradation in vitro and in animal studies, resulting in insulin resistance in adipocytes and skeletal muscle [58].

To this date, three types of receptors for chemerin have been described: chemerin chemokine-like receptor 1 (CMKLR1), C-C chemokine receptor-like 2 (CCRL2), and G protein-coupled receptor 1 (GPR1). CMKLR1 mRNA has been detected in numerous tissues such as hematopoietic, adipose tissue, endothelial cells, osteoclasts, ovarian cells. The stimulation of this receptor has the effect of enhancing leukocyte chemotaxis [64]. CCRL2 has been detected on cells such as macrophages, mast cells, lung endothelial cells, ovarian cells, lymph nodes [65]. Both of these receptors demonstrate an association with immune cells. Such an association has not been observed for GPR1 [58]. However, it is believed to be present in central nervous system cells, murine brown adipose tissue, white adipose tissue, and skeletal muscle, but its highest expression has been found in blood vessels of white subcutaneous tissue [64].

Chemerin plays a regulatory role in adipocyte differentiation and the control of proinflammatory cytokines (IL-1β, TNF-α, IL-6) in adipocytes [54,65]. Impairment to chemerin receptors in mice resulted in decreased glucose tolerance, adiposity, and reduced proinflammatory cytokine levels (TNF-α and IL-6) in adipose tissue. It confirms chemerin’s role as a molecule regulating adipose tissue development, glucose homeostasis, and inflammation [65]. It is also able to stimulate vasoconstriction, and it has been observed that suppression of its gene expression causes a decrease in blood pressure [66].

A transient phenomenon of insulin resistance is observed during pregnancy in women who were healthy before conception. In most of them, this is insignificant, and they remain euglycemic, but some women can develop glucose intolerance as GDM [59]. It has been noted that chemerin secretion during pregnancy increases significantly with the length of its duration [56]. It is not known, however, how it is related to the regulation of physiological gestation. Chemerin mRNA expression has been demonstrated in the human placenta, particularly in stromal cells and extravillous trophoblast cells, but its levels are lower compared to adipose tissue and the liver. It appears that chemerin levels are elevated during decidualization and may influence NK cell accumulation and vascular remodeling in early pregnancy, which seems to be essential during placentation [58]. However, it may play an important role in the pathogenesis of various pregnancy disorders, such as GDM, PE, intrauterine growth restriction (IUGR) [67].

The review by Bellos et al. collected 11 studies on chemerin and its concentrations during physiological pregnancy and pregnancy complicated by GDM [68]. In five of them, a significant increase in chemerin levels was observed in patients with GDM. In one of them, there was a decrease in chemerin levels, and in the remaining five, no significant association was seen.

Yang et al. conducted a study in which they compared chemerin levels in women with and without GDM during the first and third trimesters [69]. They noted a significant increase in chemerin in women with GDM in the third trimester compared to healthy women. Interestingly, chemerin levels during the first trimester were higher in women without GDM. This could explain why, in some studies, chemerin levels were lower in women with GDM compared to healthy ones.

Other studies that showed an association between elevated plasma chemerin levels and GDM were conducted by Liang et al. and Wang et al. [62,67]. Furthermore, in addition to examining plasma chemerin, Liang et al. also measured chemerin levels in other tissues such as the placenta, subcutaneous tissue, and umbilical cord blood. Each of these samples showed a positive correlation between elevated levels of chemerin and the occurrence of GDM. Wang et al. estimated the predictive value of chemerin as a marker of GDM at 73.33% sensitivity and 76% specificity. A positive correlation was also demonstrated by Okten et al., who found that salivary chemerin levels were significantly higher in women with GDM than in healthy patients [61]. Ustabay et al. evaluated the correlation between chemerin levels in breast milk and GDM [54]. Milk and serum samples were collected three times: shortly after birth—1–5 days (colostrum), 7–10 days (transitional milk), and 15–17 days (mature milk). According to them, there is a positive correlation between chemerin presented in milk and the occurrence of GDM.

Zhou et al. conducted a meta-analysis in which they assessed 11 studies on the correlation between circulating chemerin levels and the occurrence of GDM [63]. A total of 742 GDM patients and 840 healthy controls were evaluated. When comparing chemerin levels in each trimester of pregnancy, the second trimester showed the most significant increase in chemerin compared to healthy women. Recent studies have shown that adipokines, including chemerin, are released from plasma albumin. It is thought that albumin levels tend to drop with advanced pregnancy due to the increased nutritional demands of the fetus, which could partly explain the lower chemerin levels in the third trimester compared to the second one.

On the other hand, a meta-analysis conducted by Sun et al. evaluated 20 studies investigating the importance of chemerin in GDM [53]. According to their results, there was no significant difference between the levels of plasma chemerin in GDM patients compared to healthy controls.

The use of chemerin levels in assessing the effectiveness of treatment for patients with GDM was also considered. The purpose of this was to predict whether patients with GDM would need pharmacological treatment or whether dietary therapy would be sufficient. However, there was no correlation between chemerin levels and poor glucose monitoring in these patients [56].

Tsiotra et al., on the other hand, compared chemerin concentrations in serum, placenta, subcutaneous, and visceral adipose tissue in women with GDM and healthy women categorized into obese and non-obese [59]. The result of the study showed a significant difference between chemerin levels in healthy non-obese women and women with GDM and healthy women with obesity, suggesting a correlation between chemerin levels and BMI.

To demonstrate a definite link between chemerin and GDM occurrence, detailed studies of plasma chemerin levels in women with GDM need to be conducted and compared with the results of healthy patients. At this point, such studies have been performed, but the number of research is relatively small, and each has been carried out on a small sample size. Moreover, there is a lack of larger-scale studies that consider different ethnic populations.

### 2.4. Omentin-1

Omentin-1 is a member of the adipokine group, and its expression has been shown primarily in the small and large intestine, where it appears to play a role in regulating the organism’s microbiota [70]. Its expression has also been detected in the visceral adipose tissue, placenta, heart, lungs, and ovaries [56]. One of its functions is an anti-inflammatory effect, which it seems to accomplish by reducing C-reactive protein (CRP) and TNF-α levels [57]. It appears to lead to relaxation of vascular tone and vasodilation by the same mechanism. It also contributes to this by affecting endothelial nitric oxide secretion [71]. Moreover, omentin-1 impaired secretion is thought to have a contribution to the pathogenesis of coronary artery disease [59].

It is suggested that its disfunction disrupts glucose homeostasis, induces the development of insulin resistance, and thus contributes to the development of diabetes [72]. Omentin-1 is thought to be able to increase glucose uptake in adipocytes and has insulin sensitivity-enhancing abilities. It has been noted that its levels are reduced in patients with obesity and diabetes [73]. Interestingly, according to some studies, it is possible to regulate omentin-1 levels in these patients. It has been observed that weight loss can increase levels of this adipokine in obese patients with T2DM [74]. It has also been noted that omentin-1 levels are increased in pre-diabetic states, suggesting that this adipokine may increase insulin sensitivity, and its secretion is increased as a defense mechanism in the early stages of glucose intolerance [75].

In a study comparing omentin-1 levels in pregnant and non-pregnant women, it was noted that omentin-1 concentrations are significantly reduced during late pregnancy [76]. It is well known that with the duration of pregnancy, insulin resistance increases. It suggests that omentin-1, which seems to be related to the phenomenon of insulin resistance, could serve as an indicator of its occurrence [59]. Moreover, there was also observed a significant decrease in omentin-1 levels in obese women, independently of the presence of glucose intolerance [57].

To date, no certain confirmation has been found on the association between omentin-1 levels and the risk of GDM occurrence. Bellos et al. evaluated five studies on omentin-1 concentrations in pregnant women and their association with GDM in a review paper [68]. In three of these studies, omentin levels were found to be decreased in patients with GDM, while in the other two, no significant differences were observed. Importantly, in two of the studies in which patients were classified according to their BMI, significantly reduced omentin-1 levels were observed in non-obese women with GDM.

In a study performed by Tsiotra et al., it was observed that circulating omentin-1 levels were significantly reduced in GDM patients (obese and non-obese) compared to healthy controls [59]. In addition, a meta-analysis by Sun et al. conducted on 20 studies on adipokines levels and risk of GDM occurrence showed similar outcomes. According to their study, omentin-1 concentrations were significantly reduced in GDM patients [53]. They also noted that although omentin appears to be secreted primarily by visceral adipose tissue, their analysis showed a negative correlation between omentin-1 levels and the amount of visceral adipose tissue. They suggested that a decrease in omentin-1 secreted by visceral adipose tissue may trigger insulin resistance, which over time causes GDM. In their opinion, omentin-1 may serve as a predictive marker for GDM in the future.

Pan et al. conducted a detailed meta-analysis in which they examined 42 studies on the effect of omentin-1 levels on the development of different types of diabetes [65]. In 32 of these studies, the levels of this adipokine were significantly reduced in patients with T2DM compared to healthy subjects. Moreover, 7 of the studies comparing omentin-1 levels in patients with GDM and healthy controls suggested similar findings. Omentin-1 concentrations were significantly lower in women with GDM. However, these studies showed remarkably high heterogeneity, indicating that some non-described factors might have been responsible for those outcomes.

According to the Franz et al. study, omentin-1 levels did not differ between women with and without GDM [77]. In both groups, a decrease in omentin levels was observed. However, women with a higher BMI had lower omentin-1 concentrations. The offspring of women with GDM also had significantly lower levels than the offspring of healthy controls. The Francis et al. study also found no significant association between omentin-1concentrations and risk of GDM occurrence [63].

### 2.5. Osteocalcin

Osteocalcin (OC) is a non-collagenous peptide produced by bone tissue and appears to be involved in bone metabolism. Moreover, its role in glucose metabolism and hormone regulation has also been suggested. It is mainly produced by osteoblasts and odontoblasts. This molecule can be detected in small amounts in blood and used as a marker of osteogenesis [78]. Three forms of osteocalcin: carboxylated (cOC), under-carboxylated (ucOC), total osteocalcin (tOC) may be distinguished [79]. cOC seems to be involved in osteogenesis and influences osteoclast maturation. In turn, ucOC is a molecule that has a part in the regulation of the body’s glucose metabolism [80].

Its involvement in the regulation of glucose homeostasis is thought to be due to its effect on pancreatic beta cell proliferation and insulin secretion. Moreover, it also seems to increase the insulin sensitivity of cells (liver, muscle, adipose tissue) by regulating the expression of adiponectin in adipocytes. Adiponectin is one of the molecules that have the ability to sensitize tissues to insulin [81]. Studies in mice showed that lower concentrations of OC caused decreased proliferation of pancreatic beta cells, resulting in reduced insulin secretion and development of insulin resistance [82].

It appears that OC levels may be associated with metabolic syndrome, cardiovascular disease, and T2DM, but the mechanism for this association is still unknown. Some researchers believe that decreased levels of OC may be involved in the development of T2DM [80]. It also appears that ucOC levels increase significantly in response to significant weight loss, as well as its decreased insulin resistance [81]. Because OC is metabolized and removed by the kidneys, it has been noted that its levels rise notably in patients with kidney failure [83].

However, there are only a few studies evaluating OC levels in pregnant women with GDM. The mechanism causing the difference in OC concentrations between pregnant and non-pregnant women has not been discovered yet. It seems that one of the possible reasons why OC is higher in women with GDM is that there is a transient phenomenon of insulin resistance during pregnancy [84]. Insulin resistance significantly increases insulin secretion, which in turn influences bone metabolism via insulin-like growth factor 1 (IGF-1). Thus, insulin could regulate bone metabolism in pregnant women and possibly increase OC levels in women with GDM [85].

A meta-analysis by Sun et al. included seven studies of plasma OC concentrations in pregnant women [79]. A total of 628 had GDM, and 612 were healthy. They observed that ucOC and tOC levels were significantly increased in women with GDM compared to healthy controls. The authors also suggested that this was due to transient insulin resistance during pregnancy and increased secretion of IGF-1.

Winhofer et al. were the first to investigate the relationship between osteocalcin and the development of GDM [85]. They observed a significant increase in OC concentration in women with GDM between 24 and 28 gestational weeks and then 33–38 gestational weeks. Papastefanou et al. conducted a study in order to evaluate whether OC levels in women between 11 and 14 gestational weeks can predict the development of GDM in the later stages of pregnancy [86]. They found that higher OC concentrations in women in early pregnancy preceded the development of GDM in the later stages of gestation. They also combined OC with maternal and pregnancy characteristics and found a predictive algorithm with 72.2% sensitivity for 25% false-positive rates.

Martinez-Portilla et al. performed a meta-analysis of five studies investigating OC levels in women with GDM [84]. According to their findings, there was no significant difference between tOC levels in women with GDM and healthy controls during the second trimester of pregnancy. However, ucOC levels were found to be significantly higher in women with GDM compared to healthy controls. They noted that tOC levels can be influenced by the UV index and showed an inverse correlation between the mean difference in tOC and the level of UV exposure in rats. According to them, ucOC measurements could be used as a potential predictive marker for GDM. tOC does not seem to be useful in this case as it changes its concentration due to UV exposure.

There is also evidence of no association between OC levels and the occurrence of GDM. A recent study by Zhang et al. included 105 women with GDM and 46 healthy controls [83]. However, they found no significant correlation between OC levels and the occurrence of GDM. Similar conclusions were reached earlier in a study conducted by R. Saucedo et al. [78].

### 2.6. Resistin

Resistin belongs to the group of proinflammatory adipokines. Its name originates from the expression “resistance to insulin” due to its ability to inhibit the effect of insulin in mice [58]. The receptor on which resistin acts has not yet been identified. However, it has been noted that some of its proinflammatory properties are mediated through toll-like receptor 4 (TLR4) [70].

Resistin is produced in adipocytes, muscles, mononuclear cells, macrophages, bone marrow cells, and placenta [58,87]. It also seems to be expressed in pancreatic islets, which could indicate that it is involved in regulating pancreatic beta cell activity [57]. It has been noted to affect inflammatory cytokine production and monocyte-endothelial cell adhesion [56,60]. Furthermore, it seems to induce the expression of TNF-α and IL-12 in macrophages and influence the transformation of macrophages into foam cells [60].

Resistin is thought to play a significant role in the development of insulin resistance and is associated with the development of obesity, diabetes, inflammation, and metabolic syndrome [88]. This might be possible due to its role in inhibiting glucose uptake by decreasing the cell surface glucose transporter [58,87]. It increases plasma glucose concentrations in this way and thus, decreases insulin sensitivity, which in turn has a role in the pathogenesis of GDM [56]. It has also been observed that its proinflammatory properties may contribute to a decrease in the number of pancreatic beta cells [89]. Moreover, resistin has a direct effect on the increase in reactive oxygen species (ROS) production and may induce oxidative stress. As we know, oxidative stress also seems to be involved in the pathogenesis of GDM. It suggests that resistin may influence the development of this disease in more than one way [87].

Interestingly, resistin expression was detected in the placenta. Its concentration increases with the duration of pregnancy, and its maximum level is reached around full-term gestation. Its mRNA was also detected in syncytiotrophoblast in both early and late pregnancy [58]. It seems that both resistin levels and the expression of resistin mRNA are increased in the third trimester, possibly related to decreased insulin sensitivity in pregnant women at this time of gestation [57]. Some believe that resistin during pregnancy may increase insulin resistance and cause postprandial hyperglycemia, thus contributing to the development of GDM [88].

Shang et al. studied how resistin levels change in women suffering from GDM [87]. They examined maternal blood, cord blood, and placenta samples. What they found was a positive correlation between resistin levels in all sample types in women with GDM compared to healthy controls. Similar conclusions were reached by Chen et al. and Bawah et al. [90,91]. Chen et al. were the first to demonstrate an association between elevated resistin levels and GDM. Bawah et al. compared resistin levels in pregnant women between 11 and 13 gestational weeks and found that resistin concentrations were higher in those women who developed GDM at later stages of the pregnancy compared to healthy patients. A meta-analysis by Hu et al. also seems to support the conclusion that higher resistin levels are associated with a higher risk of GDM during pregnancy [92].

Gürlek et al. studied resistin levels in saliva and serum of pregnant women [89]. In both types of samples, they found higher resistin concentrations in women who later developed GDM compared to healthy controls. This suggests that both saliva and serum could predict the onset of GDM with equal reliability.

Bellos et al. conducted a review of 29 studies evaluating resistin levels in GDM [68]. Nine of these studies found a significant increase in resistin in women with GDM, while a decrease was seen in four of them. The remaining 16 studies showed no association.

Moreover, a meta-analysis conducted by Lobo et al. found no significant association between resistin levels and the occurrence of GDM in pregnancy [93]. According to them, resistin levels were similar in women with and without GDM.

### 2.7. Visfatin

Visfatin is one more of the molecules belonging to the group of novel adipokines. Earlier, this protein was known under two names: pre-B cell colony-enhancing factor (PBEF) and nicotinamide phosphoribosyltransferase (NAMPT) [94]. It is produced primarily by visceral adipose tissue, but its expression is also shown by the placenta, fetal membranes, and myometrium. It has also been observed to be secreted by the heart, lungs, kidneys, liver, muscles, bone marrow, spleen, pancreas, and brain [56,57,58,91,95].

Two isoforms of NAMPT are distinguished in the literature. These are the intracellular and extracellular forms: iNAMPT and eNAMPT [94]. The iNAMPT form is thought to be responsible for the regulation of intracellular levels of oxidized nicotinamide adenine dinucleotide (NAD+) and has an enzymatic role. It appears to be an important factor in free radical production, cell adhesion, aging, and longevity [58,60,95]. The eNAMPT form, on the other hand, seems to have a role similar to cytokines and may be responsible for inflammation and cellular stress. It is thought to have insulin-mimetic effects [94]. It is believed to be produced by a variety of cells, such as adipocytes, hepatocytes, myocytes, and pancreatic cells [95]. It may also bind TLR4, activate its signaling pathway, and induce the development of inflammatory response [96].

Interestingly, eNAMPT has been observed to be secreted in vitro by melanoma cells, suggesting its use as a potential marker of carcinogenesis [97]. To date, the receptor by which visfatin interacts has not been discovered, and the underlying mechanism by which it has an extracellular effect is unknown [58].

Visfatin is also produced by macrophages and neutrophils. One of its possible working mechanisms is thought to be the production of proinflammatory cytokines such as IL-1, IL-6, TNF-α. Moreover, it also seems to induce leukocyte chemotaxis [57,98].

Visfatin is thought to play an important role in initiating the inflammatory response and thus contributes to the development of insulin resistance and obesity [99]. It appears to do this via the nuclear factor kappa-light-chain-enhancer of activated B cells (NF-κB) pathway. According to some studies, one of its important functions is to regulate energy homeostasis in the organism, and its release is increased in states of hyperglycemia. It seems that it may be associated with increased BMI, T2DM, and metabolic syndrome [58,60,95]. It has been observed that visfatin expression may be dependent on insulin-sensitizing factors, while visfatin itself has insulin-mimetic abilities and can bind to the insulin receptor. These abilities would be to reduce hepatic glucose release and stimulate glucose usage in myocytes and adipocytes [57,59]. Interestingly, in a study conducted on patients with polycystic ovary syndrome (PCOS), a significant reduction in visfatin levels was noted after 3 months of metformin therapy [58].

An increase in visfatin mRNA expression in subcutaneous tissue was observed during pregnancy [57]. It has also been found in fetal membranes, mesenchymal cells, chorionic cytotrophoblast, and parietal decidua [95]. It seems to reach its peak values between 19 and 26 gestational weeks [56]. Increased BMI in the pre-pregnancy period and obesity during pregnancy may trigger an inflammatory response and thus lead to the development of obstetric pathologies. According to some sources, visfatin levels increase, while other researchers believe the opposite. However, it seems to be significant in the pathogenesis of PE, IUGR, and preterm labor [58,95]. Its mechanism of inducing labor would be based on stimulation of IL-6 and IL-8 secretion. Moreover, the effect of phospholipid metabolism in the placenta through activation of the NF-κB pathway may also be important [58,95]. It seems to play a role in the regulation of fetal growth. A positive correlation of its concentration with neonatal birth weight has been observed by Mazaki-Tovi et al. [100]. Taraqi et al., in turn, found a negative correlation between visfatin levels and infants’ birth weight whose mothers had a BMI higher than normal [101].

Bellos et al. prepared a review of 29 studies that evaluated visfatin levels in women with GDM [68]. Of these, seven confirmed its increased levels in women with GDM compared to healthy ones. Another 7 studies showed decreased visfatin levels in women with GDM, while the remaining 15 did not show any association. O’Malley et al., in the latest study, confirmed the lack of association between visfatin levels and the risk of GDM occurrence. The study included 105 women with GDM and 91 healthy controls, and visfatin values were found to be similar in both groups [102].

Two of the studies that considered BMI values showed significant differences in visfatin levels in obese women. Some believe that visfatin is not so much related to the occurrence of GDM but rather to BMI itself as the most important factor [68]. Similar conclusions were reached by Zhang et al. in their meta-analysis [103]. They observed a significant increase in visfatin in obese women with GDM; however, they pointed out that it was obesity that played the most important role.

Varma et al. demonstrated a positive correlation between visfatin expression in visceral and subcutaneous adipose tissue and BMI [104]. Ferreira et al. observed increased visfatin levels during the first trimester in women who later developed GDM [105]. Similar conclusions were reached by Bawah et al. [91]. Their study investigated visfatin levels in women in their first trimester of pregnancy (11–13 gestational weeks). They observed that visfatin levels were higher in women who later developed GDM. They estimated the sensitivity of visfatin as a potential predictive marker of GDM at 87.1% and the specificity at 70%.

On the other hand, Tsiotra et al. found that visfatin levels were significantly lower in women with GDM and obesity compared to healthy, non-obese women [91]. A similar association was suggested by Akturk et al. [106]. According to their study, which measured visfatin in pregnant women between 33 and 39 gestational weeks, levels of this molecule were found to be lower in women who developed GDM compared to healthy controls.

### 2.8. Vaspin

Visceral adipose tissue-derived serine protease inhibitor (vaspin) is a member of the serine enzyme inhibitor family [107]. It was first identified in visceral adipose tissue of Otsuka Long-Evans Tokushima Fatty (OLETF) rats, which is characterized by central obesity and T2DM [108]. Studies have shown that vaspin mRNA expression and serum levels correlated positively with obesity and insulin resistance and decreased with the development of diabetes. This is presumed to be a compensatory mechanism in severe insulin resistance [109].

Vaspin was a novel adipocytokine that was mainly expressed in visceral adipose tissue, but researchers have also demonstrated its presence in the placenta [110]. Its concentration increases gradually, reaching the highest concentration at the end of pregnancy. This may mean that an increase in vaspin concentration is associated with fetal growth [111]. Giomisi et al. postulated that vaspin may act as a protective cytokine in GDM [112]. However, the role of vaspin in the pathogenesis of GDM remains poorly understood, and the correlation between vaspin levels and the development of GDM remains somewhat controversial.

Unfortunately, there is no consistency in the results of recent studies [68]. The most current of these indicates elevated serum vaspin levels in women with GDM. In this study, researchers investigated changes in serum vaspin levels in pregnant women with GDM after glucose loading during OGTT. The study involved 30 patients with GDM and 30 age-matched pregnant women with normal glucose tolerance (NGT) [113]. Liu et al. also analyzed the effect of blood glucose on serum vaspin secretion in pregnant women with GDM [84]. The results of the study indicate that during OGTT, vaspin concentration was higher in women in the GDM group than in the NGT group; furthermore, after 1 and 2 h of the test, vaspin concentration was significantly higher than baseline vaspin concentration [113].

This suggests that vaspin concentration in the GDM group may be regulated by the level of hyperglycaemia. In addition, the change in vaspin concentration positively correlated with blood glucose and lipid levels in GDM women, especially after 1 h OGTT, which was not observed in the NGT group. This means that it may be involved in the pathogenesis of GDM and related to lipid metabolism [84]. Similar findings were obtained by Tang et al. [114]. They showed that serum vaspin levels and vaspin expression in adipose tissue were significantly higher in pregnant women with GDM than in controls [85]. However, the underlying mechanism of this phenomenon needs to be further investigated. In contrast, other studies on vaspin show that its serum levels were significantly lower in the GDM group than in the control group [115]. Huo et al. analyzed serum vaspin levels and its expression in the placenta in 30 pregnant women with GDM. The control group included 27 women [115]. The serum vaspin concentration of women in the GDM group during pregnancy was significantly decreased compared with the control group. Only vaspin expression in the placenta of women with GDM was described, but there was no difference between the GDM group and the control group [115]. Mierzyński et al. determined vaspin levels in 153 women with GDM and in 84 pregnant women with normal glucose tolerance [116]. In the study, patients with GDM had lower vaspin levels compared to the NGT group [116].

In summary, the conflicting nature of the outcomes regarding vaspin precludes the draw of a safe conclusion. However, recent studies suggest that it may be involved in the pathogenesis of GDM and related to lipid metabolism. Nonetheless, the underlying mechanism needs to be further investigated.

### 2.9. Irisin

Irisin is a novel myokine [117], adipokine [118], and neurokine [119], which is mainly secreted by skeletal muscle. During muscle contraction, irisin is released into the bloodstream through the stimulation of peroxisome proliferator-activated receptor gamma coactivator 1-alpha (PGC-1α) and proteolytic cleavage of fibronectin type III domain-containing protein 5 (FNDC5) [117]. Irisin is expressed in the female reproductive system, including the ovary, as well as in the placenta [120].

Irisin plays an important role in fat metabolism and energy homeostasis by mediating exercise-related energy expenditure. This process is accomplished by converting white adipose tissue into brown adipose tissue in response to activation of PGC-1 [117]. This leads to an increase in total energy expenditure, which is associated with weight loss. Irisin also increases glucose tolerance and insulin sensitivity. To date, several studies have shown that irisin plays a significant role in metabolic diseases. These may include obesity [121], T2DM [122], lipid metabolism [123], cardiovascular disease [124], nonalcoholic fatty liver disease [125] and polycystic ovary syndrome [126]. Given these findings, the association between GDM and irisin levels has become an interesting subject of many studies in recent years.

Kulhan et al. evaluated the association between serum irisin levels and GDM. They also tried to estimate the possible benefits of the metabolic profile [127]. The results of the study showed that levels of serum irisin were lower in pregnant women with GDM compared to women with uncomplicated pregnancies [127]. In this cross-sectional study, the researchers noted that serum irisin levels in pregnant women were negatively correlated with BMI but positively correlated with insulin resistance. Kulhan et al. [127] hypothesized that treatments aimed at increasing irisin levels may be more beneficial before the development of GDM to prevent complications that may occur in pregnancy. However, this study has some limitations because irisin was detected in patients already diagnosed with GDM. The researchers did not prove that irisin could be a predictive marker of risk for developing GDM in later pregnancy.

AL-Ghazali et al. [128] also confirmed that serum irisin levels in pregnant women with GDM are significantly lower compared to healthy pregnant women. These results, according to the authors of the study, may suggest compensation for the physiological increase in insulin resistance or a stimulatory effect of high estrogen concentrations [128]. It also takes into account possibly its additional secretion by the placenta, although the effect of placental tissue on circulating irisin appears to be insignificant [129]. However, the present results suggest that a low level of serum irisin may be a novel marker for GDM.

Similar results were reported in a meta-analysis by Cui et al. [130]. They found that low irisin levels may contribute to increased serum glucose and decreased insulin sensitivity. Furthermore, this study found low irisin levels not only in maternal blood but also in cord blood and breast milk [130]. Fatima et al. [131] believe that continuous breast milk feeding with lower concentrations of irisin may negatively affect health and lipid regulation. However, the relationship between low levels of irisin in breast milk and infant weight requires further study.

Irisin may play a key role in the development of GDM. Recent studies show significantly lower irisin levels in women with GDM compared to women with normal pregnancies as controls, which may be a promising marker for the diagnosis of GDM in the future. However, as of today, we need more studies on this subject.

### 2.10. Apelin

Apelin is a bioactive peptide that was first isolated from bovine stomach tissue. It has been characterized as an endogenous ligand of the receptor APJ, which belongs to the G protein-coupled receptor family [132]. In 2005, it was described as a novel adipokine that is secreted by mature human adipocytes [133]. Apelin and its receptor are widely expressed in various types of central and peripheral tissues, particularly in the cardiovascular system, central nervous system, reproductive system, including ovary and placenta. It has been studied that apelin is an adipokine involved in glucose homeostasis, and its levels are increased in obesity and T2DM [134,135]. However, its role in the pathogenesis of GDM has been poorly described, and the results of available studies on this subject are inconsistent.

Apelin levels in patients with GDM are higher than in women with healthy pregnancies [136,137]. Moreover, this level increases during pregnancy and reaches its highest value in the third semester [136]. In contrast to the results of the above studies, other researchers noted that the level of apelin in peripheral blood was lower in patients with GDM [138]. Moreover, this is confirmed by the results of two reliable review studies, which suggest that there are no differences in apelin levels in GDM and a control group [68]. Apelin mRNA expression in the placenta was significantly higher than in adipose tissue [53,139,140]. However, apelin mRNA expression in placental tissue samples in women with GDM was not altered compared with healthy individuals [139]. High placental apelin expression may imply that its fetoplacental activity is mediated in a paracrine or autocrine way; unfortunately, the specific signals that regulate its availability are still unclear [139].

In conclusion, studies on apelin and its association with the pathophysiology of GDM are inconsistent. According to the current state of knowledge, it is unlikely that apelin can be a diagnostic marker in GDM in the future.

### 2.11. FABP4

Fatty acid-binding proteins are a family involved in lipid metabolism. FABP4 was first identified in adipose tissue and mature adipocytes [141]. It is a relatively novel adipokine [142]. This protein is also termed adipocyte P2 (aP2) or adipocyte FABP (A-FABP) [143]. FABP4 is highly expressed during adipocyte differentiation and transcriptionally controlled by peroxisome proliferator-activated receptor γ (PPARγ) agonists, fatty acids (FAs), dexamethasone, and insulin [144]. It represents approximately 1% of all soluble proteins in adipose tissue [143]. FABP4 expression is also induced during the differentiation of monocytes into macrophages, and its expression in these cells is regulated by a wide range of proinflammatory stimuli [144].

FABP4 secretion from adipocytes occurs under the influence of lipolytic agonists or nutrient deprivation. This is probably due to the control of glucose production by hepatocytes and insulin secretion by pancreatic β- cells [145]. Glucose oxidation and glycolysis are inhibited, and glucose uptake and use in muscle and liver are significantly reduced. Direct effects of exogenous FABP4 were demonstrated in vascular endothelial cells, where it inhibited the expression or activation of endothelial nitric oxide synthase (eNOS). Furthermore, in cardiomyocytes, it reduced contraction [144,146].

It has been proven that there is an association between FABP4 and obesity markers such as BMI and body fat levels. Elevated FABP4 levels positively correlate with obesity-related diseases, T2DM, including polycystic ovary syndrome. The effect of this adipokine on the correlation between obesity and insulin resistance has also been noted. Moreover, serum FABP4 levels were shown to be significantly the highest in the GDM group in the early puerperium, which may suggest that elevated FABP4 levels may persist in GDM patients postpartum and contribute to increased risk of T2DM and metabolic syndrome [147]. It is highly likely that FABP4 is involved in the pathophysiology of GDM [148].

In previous studies, researchers have observed elevated serum FABP4 levels in women diagnosed with GDM compared to healthy controls [67,79,149,150,151,152]. Fancis et al. [65] evaluated the concentrations of a panel of 10 adipokines during pregnancy and their association with GDM risk. The study found that higher FABP4 levels in early and middle pregnancy were significantly associated with an increased risk of GDM. Throughout the pregnancy, FABP4 levels were higher among women diagnosed with GDM compared with controls [65]. Zhang et al. [151] also demonstrated that serum FABP4 levels in patients with GDM increased continuously from the second to the third trimester. Ortega-Senovilla et al. [153] found that serum FABP4 levels observed in women with GDM were higher than in controls when FABP4 values were adjusted for pre-pregnancy BMI. In addition, it was reported that FABP4 levels were significantly higher in the GDM group [153]. Moreover, FABP4 is an independent risk factor for increased insulin resistance in pregnancy [154]. Dong et al. [155] provided further evidence of a correlation between FABP4 and insulin resistance in GDM. In addition, they investigate the effects of the FABP4 inhibitor BMS309403 on GDM mice. Inhibition of FABP4 by BMS309403 resulted in significant relief of GDM symptoms in the GDM mouse model, including improved glucose and insulin sensitivity. Researchers proposed that it may be an effective treatment for alleviating GDM [150].

High levels of circulating FABP4 in the serum of pregnant women with GDM may be caused by the additional release of this adipokine from the placenta and adipocytes [155]. It is supposed that overexpression of FABP4 in the placenta may be induced by human placental lactogen and progesterone. Since the concentration of these hormones increases steadily throughout pregnancy, it is thought that it may be associated with an increase in insulin resistance with advancing gestational age [155]. Synergistic release of FABP4 from the placenta and adipocytes may affect both metabolic and inflammatory pathways. This may play a key role in the future development of T2DM in post-partum women [155].

Considering the previous studies, it seems that FABP4 can be used as a predictive marker in the diagnosis of GDM. This is supported by a few of the most recent meta-analyses, which found FABP4 to be the most promising predictor of GDM [68,156].

### 2.12. FGF21

FGF21 belongs to the FGF superfamily with a wide range of biological functions. They participate in cell proliferation and differentiation, neuronal development, angiogenesis, and various metabolic processes [157]. FGF21 is mainly secreted by the liver, but it is also expressed in other metabolically active tissues such as the pancreas, skeletal muscle, adipose tissue, including the placenta [158,159]. It is involved in the regulation of glucose and lipid metabolism. FGF21 also has been shown to be an important regulator in maintaining energy homeostasis. Previous studies have shown that FGF21 can improve insulin resistance in peripheral tissues of patients with T2DM, leading to decrease glucose levels independent of insulin [160]. This raises the suspicion that FGF21 may also be involved in the pathophysiology of GDM.

There are a limited number of studies examining FGF21 levels in patients with GDM. In addition, the results obtained are inconsistent. Previous studies have found higher levels of circulating FGF21 in pregnant women with GDM than in control women. Wang et al. [161] demonstrated that serum FGF21 levels were increased in patients with GDM. Additionally, they noted that serum FGF21 levels positively correlated with insulin resistance and serum triglyceride levels. They hypothesized that increased serum FGF21 levels may be a compensatory response to this disease [161]. The elevation of FGF21 levels in GDM is consistent with the results of Tan et al. [162]. They also examined FGF21 levels in cerebrospinal fluid (CSF). However, there was no significant difference in FGF21 concentration in CSF in women with GDM compared with controls. FGF21 has been detected in human cord blood. It was observed that lower levels of FGF21 in cord blood were strongly positively correlated with its levels in the maternal circulation [162]. Megia et al. [163] described that FGF21 levels in cord blood were correlated with infant BMI at 12 and 24 months of age, and this correlation was stronger in the group with normal glucose tolerance. This may suggest a potential role for FGF21 in intrauterine life that may influence future metabolic disorders [163].

In contrast to the above studies, Mosavat et al. [164] showed that FGF21 was lower in the GDM group compared to healthy pregnant women. Serum FGF21 levels were associated with metabolic risk factors in pregnancy, including insulin resistance, elevated TG levels, and decreased HDL. However, it may not be a major factor in the pathogenesis of GDM [164]. In addition, reduced serum FGF21 levels in GDM patients were shown by Xu et al. [165]. However, a major limitation of this study is that the control group was not pregnant, so it cannot be compared to previous studies.

Although understanding of the role of FGF21 in T2DM has increased over the past decade, its role in the pathophysiology of GDM remains unclear. Further studies are needed on the relationship between this adipokine and GDM.

### 2.13. Lipocalin-2

Novel marker predictors of diabetes mellitus have been the subject of many studies over the years. Lipocalin-2, built from 178 amino acids, also named neutrophil gelatinase-associated lipocalin (NGAL) or oncogene 24p3, belongs to a family of small binding proteins that may be produced by the immune system [166]. This 25 kDa glycoprotein identified as a cytokine is expressed in multiple tissues, mainly in trophoblasts and adipocytes [167]. It was first discovered in human neutrophils [168]. In low levels, it is secreted in the kidney, prostate, liver, spleen, uterus, salivary gland, stomach, colon bone marrow, and epithelia of the respiratory and alimentary tracts [169]. Lipocalin-2 is also found in secretions such as colostrum, amniotic fluid, and cervical mucus [170]. NGAL may bind both steroids and lipopolysaccharides. Its role is to co-operate apoptosis and metabolism, transport fatty acids, antibacterial protection, embryogenesis, and regulation of iron level [171]. The solute carrier family 22 member 17 and the megalin/glycoprotein GP330 are described to be two receptors for lipocalin-2 [172]. Elevated levels of lipocalin-2 may be associated with obesity, hypertriglyceridemia, hyperglycemia, markers of insulin resistance, and declining pancreatic β-cell function [173].

Some studies show the role of lipocalin-2 in degenerative diseases, especially in rheumatoid arthritis. NGAL is being expressed by IL-1β, lipopolysaccharide (LPS) in the chondrocytes [174]. What is more, elevated oncogene 24p3 concentrations are found in heart failure, coronary heart disease, inflammatory bowel disease (IBD), polycystic ovary syndrome, and stroke. It promotes plaque and aneurysm formation. The level of lipocalin-2 in patients with GDM is significantly higher, especially in Caucasians with BMI > 25 kg/m^2^. South and East Asians have poor predictive ability [30,148]. It is an independent biomarker-predictor in GDM, especially in early pregnancy (9–12 weeks of gestation) [175]. Of all adipokines, lipocaine-2 has proven to be the most reliable and significant predictive factor. Upregulation of lipocalin-2 is mediated by phosphatidylinositol 3-kinase and mitogen-activated protein kinase. A higher level of NGAL is associated with higher BMI, FPG, FPI, HOMAI-IR, TG, TC, hs-CRP in women with GDM, and there is a negative correlation with HDL-C or LDL-C [176]. In one research significantly higher concentration of lipocalin-2 was found two days after the beginning of lactation in women with GDM in comparison to healthy patients [177]. A positive correlation between elevated concentrations of NGAL and developed PE was also demonstrated [178]. The second correlation was found between lipocalin-2 and TNF-α, a predictor of insulin resistance in human pregnancy [179]. NGAL has a complex biological role and is involved in the pathogenesis of many diseases. Taking into consideration the complexity of the impact on different tissues, it is still a very challenging subject. Further studies are necessary to determine the role of lipocalin-2 in pathological pregnancies affected by GDM.

## 3. Conclusions

GDM is a global problem that takes its toll with each passing year. The course of GDM is usually asymptomatic; therefore, carbohydrate disturbances can be detected while performing a blood glucose test. Impaired carbohydrate tolerance, which the patient developed for the first time during pregnancy, carries serious consequences for both the mother and her child. The future health of a pregnant woman as well as of her offspring is in her own hands. By following certain rules, e.g., forming and maintaining eating habits, taking physical exercise, watching body weight, a mother-to-be can prevent complications associated with the development of GDM.

In light of this, the crux of the matter is to find a biopredictor capable of singling out those women who are at risk of developing GDM at the very start of pregnancy. Unfortunately, the answer to the most important question: “Can we predict the onset of GDM using one reliable biomarker before the disease has developed?” for now is “Not yet”.

Most of the original scientific studies presented in this review are retrospective, which means that they were carried out in pregnant women already diagnosed with GDM. Hypotheses and pathways of these 13 less known biomolecules in the pathogenesis of GDM are presented in Figure 1.

Our study investigates the 13 biomolecules with different mechanisms of action (Table 1).

Moreover, 7 of them present controversial data concerning the serum concentrations in women with GDM compared to healthy pregnant women (Figure 2).

Considering the non-invasive nature of collecting saliva or urine samples, it seems that such biomaterials would be the most optimal factors in predicting any disease. Out of many biomolecules circulating in the blood, chemerin and resistin contained in the saliva are worth paying attention to since their levels are reported to be significantly higher in women with GDM in comparison to healthy women. The above-described observations may be suggestive of the validity of performing non-invasive tests, the use of which could detect the risk of developing GDM.

Taking into account the presented studies, it seems that

-High levels of FABP4;-High levels of one form of osteocalcin (i.e., ucOC);-Low levels of irisin in the serum of pregnant women can be used as predictive markers in the diagnosis of GDM.

Early diagnosis of GDM has a significant impact on the development of the fetus, the course of delivery, and the neonatal period. It also influences the later development of the child as well as on subsequent maternal complications. Scientific progress stimulates the advancement in research work aiming at optimization of the molecular methods, i.e., development of new diagnostic methods and improvement of the existing analytical procedures.

Currently, these molecules have no application in the treatment of GDM, which is usually based on a diet. If the diet is not sufficient, insulin or oral antidiabetic medicaments are used depending on specific national guidelines regarding the management of GDM.

Hopefully, future clinical trials will shed more light on the validity of these hypotheses and, more importantly, determine which biomolecule has the most potential to predict GDM pathogenesis.

## Figures and Tables

**Figure 1 ijms-22-11578-f001:**
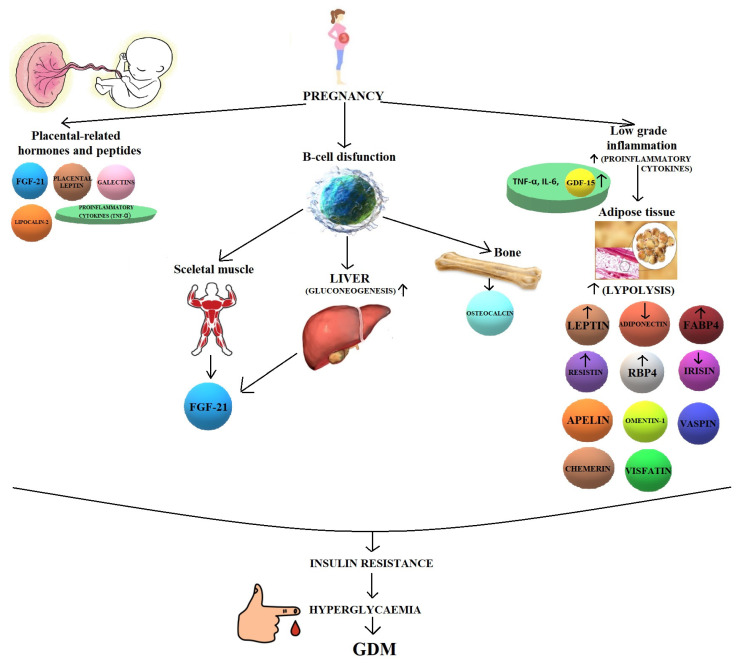
Hypotheses and pathways of novel biomolecules in the pathogenesis of GDM. FABP4, fatty acid-binding protein 4; FGF21, fibroblast growth factor 21; GDF-15, growth differentiation factor 15; GDM, gestational diabetes mellitus.

**Figure 2 ijms-22-11578-f002:**
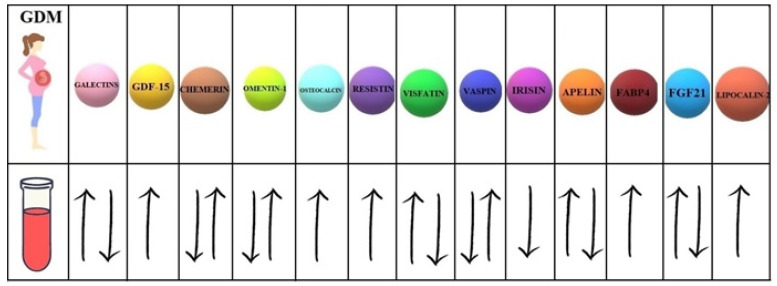
Concentrations of selected biomolecules in the serum of GDM patients compared to their concentrations in the serum of healthy pregnant women. FABP4 (fatty acid-binding protein 4); FGF21 (fibroblast growth factor 21); GDF-15 (growth differentiation factor 15).

**Table 1 ijms-22-11578-t001:** Potential mechanisms of action of selected biomolecules.

Biomolecules	Localization	Mechanism of the Action
GALECTINS	PLACENTA, ADIPOSE TISSUE, EOSINOPHILS, BASOPHILS MACROPHAGES (M1, M2), THYMUS, KIDNEY, SYNOVIAL FLUID, INTESTINE, STOMACH, MUSCLES, NEURONS, T-HELPER CELLS	Activation of apoptosis in T cells [38], Suppression of Th1 and Th17 immune responses [41], Inhibition of B cell receptor activation [39], Involvement in direct insulin secretion of pancreatic beta cells [40], Stimulation of apoptosis and cellular repair mediated by p53 [38], Stimulation of apoptosis of adipocytes [38]
VASPIN	ADIPOSE TISSUE, PLACENTA	Upregulation of the PI3-K/Akt signaling pathway and inhibition free fatty acid-induced apoptosis of vascular endothelial cells [109]
RESISTIN	ADIPOCYTES, MUSCLES, MONONUCLEAR CELLS, MACROPHAGES, BONE MARROW CELLS, PLACENTA, PANCREATIC ISLETS	Upregulation of intercellular adhesion molecule-1 (ICAM1) vascular cell-adhesion molecule-1 (VCAM1) and chemokine (C-C motif) ligand 2 (CCL2) [60], Itself can be upregulated by interleukins (ILs) and also by microbial antigens such as lipopolysaccharide [72]
VISFATIN	ADIPOSE TISSUE, PLACENTA, FETAL MEMBRANES, MYOMETRIUM, HEART, LUNGS, KIDNEYS, LIVER, MUSCLES, BONE MARROW, SPLEEN, PANCREAS, BRAIN	Upregulated by hypoxia, inflammation and hyperglycaemia/downregulated by insulin, somatostatin and statins [57,58,60,95,98], Regulation of NAD+ biosynthesis [58,60,95], Possession of both cytokine-like extrinsic activity (PBEF) and enzymatic intrinsic activity (NAMPT) [94]
GDF-15	PLACENTA, FETAL MEMBRANE, BLADDER, PROSTATE, STOMACH, DUODENUM	Secretion mediated through mitochondrial stress and by activation of the integrated stress response pathway as well as, potentially, via 5’AMP-activated protein kinase (AMPK) [49], Promotion of the polarization of macrophages to the M2 phenotype in adipose tissue [51]
OMENTIN-1	SMALL AND LARGE INTESTINE, VISCERAL ADIPOSE TISSUE, PLACENTA, HEART, LUNGS, OVARIES	Reducing C-reactive protein (CRP) and tumor necrosis factor-alpha (TNF-α) levels [51]
OSTEOCALCIN	BONE TISSUE	Increase the insulin sensitivity of cells (liver, muscle, adipose tissue) by regulating the expression of adiponectin in adipocytes [81] Decreased proliferation of pancreatic beta cells, resulting in reduced insulin secretion and development of insulin resistance [82]
APELIN	STOMACH TISSUE, ADIPOCYTES, CARDIOVASCULAR SYSTEM, CENTRAL NERVOUS SYSTEM, REPRODUCTIVE SYSTEM, OVARY, PLACENTA	Action on an AMPK-dependent mitochondria biogenesis [139]
IRISIN	SKELETAL MUSCLES, OVARY, PLACENTA, ADIPOSE TISSUE	Stimulation of peroxisome proliferator-activated receptor gamma coactivator 1-alpha (PGC-1α) and proteolytic cleavage of fibronectin type III domain-containing protein 5 (FNDC5) [117], Activation of PGC-1 [117]
FABP4	ADIPOSE TISSUE, VASCULAR AND ENDOTHELIAL CELLS, CARDIOMYOCYTES, PLACENTA	Controlled by peroxisome proliferator-activated receptor γ (PPARγ) agonists, fatty acids (FAs), dexamethasone and insulin [144], Inhibition of the expression or activation of endothelial nitric oxide synthase (eNOS) [144,146], Overexpression of FABP4 in the placenta may be induced by human placental lactogen and progesterone [155]
CHEMERIN	WHITE ADIPOSE TISSUE, LIVER, PLACENTA, LUNGS, SKELETAL MUSCLES, KIDNEYS, OVARIES, HEART, ADRENAL GLANDS	Action as a ligand for the G protein-coupled receptor—chemerin chemokine-like receptor 1 (CMKLR1) [64], Secretion from adipose tissue is enhanced by insulin [58], Stimulation of the phosphorylation of the mitogen-activated protein kinases (MAPKs), extracellular signal-regulated kinase 1 (ERK1), and ERK2, which are involved in mediating lipolysis [54,55,56,57]
LIPOCALIN-2	THROPHOBLASTS, ADIPOCYTES, NEUTROPHILES, KIDNEYS, PROSTATE, LIVER, SPLEEN, UTERUS, SALIVARY GLAND, STOMACH, COLON, BONE MARROW, EPITHELIA OF THE RESPIRATORY AND ALIMENTARY TRACTS	Expressed by IL-1β, lipopolysaccharide (LPS) in the chondrocytes [174], mediated by phosphatidylinositol 3-kinase and mitogen-activated protein kinase [176], Action as bone-derived hormone that crosses the blood brain barrier and acts on paraventricular nucleus of hypothalamus in the brain [30,153], Action as tumor protease inhibitors [30,153]
FGF21	PANCREAS, SKELETAL MUSCLES, ADIPOSE TISSUE, PLACENTA, LIVER	Induced by mitochondrial 3-hydroxy-3-methylglutaryl-CoA synthase 2 (HMGCS2) activity, a sirtuin 1 (SIRT1)-dependent mechanism [161], Regulated by PPARα, Phosphorylation of fibroblast growth factor receptor substrate 2 (FRS2), a protein linking FGF receptors to the Ras/MAP kinase pathway [158,159]

## Data Availability

MDPI Research Data Policies.

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
