# Peer review of "Novel Biomolecules in the Pathogenesis of Gestational Diabetes Mellitus"

_ijms, 2021, doi:10.3390/ijms222111578_

Round 1

Reviewer 1 Report

The article entitled "Novel biomolecules in gestational diabetes mellitus" presents the discussion relevant to various biomolecules that play critical roles in the pathogenesis of GDM. The authors discussed well and compiled the required biomolecules. However, several minor comments are to be addressed prior to its publication.

Better make the title clear in terms of roles of these biomolecules i.e., pathogenesis

Are there any reports on the treatment of GDM due to these molecules? Better add discussions in their corresponding sections.

Add citations in the table.

Author Response

Dear Reviewer 1,

            We would like to resubmit our manuscript entitled “Novel biomolecules in the pathogenesis of gestational diabetes mellitus”. We appreciate your valuable remarks and hope that the quality of our manuscript is going to meet your expectations now that we have made some suggested alternations.

“The article entitled "Novel biomolecules in gestational diabetes mellitus" presents the discussion relevant to various biomolecules that play critical roles in the pathogenesis of GDM. The authors discussed well and compiled the required biomolecules. However, several minor comments are to be addressed prior to its publication.”

             Thank you very much for finding the time to read our manuscript. Thank you for considering our manuscript. We find all your remarks spot on therefore we have made a point-by-point correction of the manuscript according to your suggestions.

“1. Better make the title clear in terms of roles of these biomolecules i.e., pathogenesis”

We have modified the title of our manuscript. It now reads: “Novel biomolecules in the pathogenesis of gestational diabetes mellitus”. We aimed at highlighting the role of these biomolecules in the pathogenesis of GDM.

“2. Are there any reports on the treatment of GDM due to these molecules? Better add discussions in their corresponding sections.”

Following your advice, we have included these sentences:

“Taking into account the presented studies, it seems that

  • high levels of FABP4,
  • high levels of one form of osteocalcin (i.e. ucOC),
  • low levels of irisin

in the serum of pregnant women can be used as predictive markers in the diagnosis of GDM.

[…]

Currently, these molecules have no application in the treatment of GDM, which is usually based on a diet. If the diet is not sufficient, insulin or oral antidiabetic medicaments are used depending on specific national guidelines regarding management of GDM.

We have also decided to add this sentence into the study abstract: “It seems that high levels of FABP4, low levels of irisin, and high levels of under-carboxylated osteocalcin in the serum of pregnant women can be used as predictive markers in the diagnosis of GDM”.

“3. Add citations in the table.”

We have added the adequate citations into the table, i.e.:

References

39.Tian, J.; Hu, S.; Wang, F.; Yang, X.; Li, Y.; Huang, C. PPARG, AGTR1, CXCL16 and LGALS2 polymorphisms are correlated with the risk for coronary heart disease. Int. J. Clin. Exp. Pathol. 2015, 8, 3138–3143.

40.Lorenzo-Almoros, A.; Hang, T.; Peiro, C.; Soriano-Guillen, L.; Egido, J.; Tunon, J.; Lorenzo, O. Predictive and diagnostic biomarkers for gestational diabetes and its associated metabolic and cardiovascular diseases. Cardiovasc. Diabetol. 2019, 18, 140.

41.Blois, S.M.; Gueuvoghlanian-Silva, B.Y.; Tirado-González, I.; Torloni, M.R.; Freitag, N.; Mattar, R.; Conrad, M.L.; Unverdorben, L.; Barrientos, G.; Knabl, J.; Toldi, G.; Molvarec, A.; Rose, M.; Markert, U.R.; Jeschke, U.; Daher, S. Getting too sweet: galectin-1 dysregulation in gestational diabetes mellitus. Mol. Hum. Reprod. 2014, 644–649.

42.Tirado-Gonzalez, I.; Freitag, N.; Barrientos, G.; Shaikly, V.; Nagaeva, O.; Strand, M.; Kjellberg, L.; Klapp, B.F.; Mincheva-Nilsson, L.; Cohen, M.; Blois, S. Galectin-1 influences trophoblast immune evasion and emerges as a predictive factor for the outcome of pregnancy. Mol. Hum. Reprod. 2013, 19, 43-53.

51.Petersen, K.F.; Befroy, D.; Dufour, S.; Dziura, J.; Ariyan, C.; Rothman, D.L.; DiPietro, L., Cline, G.W.; Shulman, G.I. Mitochondrial dysfunction in the elderly: possible role in insulin resistance. Science 300 2003, 5622, 1140–1142.

53.Lee, S.E.; Kang, S.G.; Choi, M.J.; Jung, S.B.; Ryu, M.J.; Chung, H.K.; Chang, J.Y.; Kim, Y.K.; Lee, J.H.; Kim, K.S.; Kim, H.J.; Lee, H.K.; Yi, H.S.; Shong, M. Growth differentiation factor 15 mediates systemic glucose regulatory action of T-Helper type 2 cytokines. Diabetes 2017, 11, 2774–2788.

56.Ustebay, S.; Baykus, Y.; Deniz, R.; Ugur, K.; Yavuzkir, S.; Yardim, M.; Kalayci, M.; Çaglar, M.; Aydin, S. Chemerin and Dermcidin in Human Milk and Their Alteration in Gestational Diabetes. J. Hum. Lact. 2019, 35, 550-558.

57.Kennedy, A.J.; Davenport, A.P. International Union of Basic and Clinical Pharmacology CIII: Chemerin Receptors CMKLR1 (Chemerin1) and GPR1 (Chemerin2) Nomenclature, Pharmacology, Function. Pharmacol. Rev. 2018, 70, 174-196.

58.Gutaj, P.; Sibiak, R.; Jankowski, M.; Awdi, K.; Bryl, R.; Mozdziak, P.; Kempisty, B.; Wender-Ozegowska, E. The Role of the Adipokines in the Most Common Gestational Complications. Int. J. Mol. Sci. 2020, 21.

59.de Gennaro, G.; Palla, G.; Battini, L.; Simoncini, T.; Del Prato, S.; Bertolotto, A.; Bianchi, C. The role of adipokines in the pathogenesis of gestational diabetes mellitus. Gynecol. Endocrinol. 2019, 35, 737-751.

60.Estienne, A.; Bongrani, A.; Reverchon, M.; Ramé, C.; Ducluzeau, P.H.; Froment, P.; Dupont, J. Involvement of Novel Adipokines, Chemerin, Visfatin, Resistin and Apelin in Reproductive Functions in Normal and Pathological Conditions in Humans and Animal Models. Int. J. Mol. Sci. 2019, 20.

62.Šimják, P.; Cinkajzlová, A.; Anderlová, K.; Pařízek, A.; Mráz, M.; Kršek, M.; Haluzík, M. The role of obesity and adipose tissue dysfunction in gestational diabetes mellitus. J. Endocrinol. 2018, 238, 63-77.

66.Rourke, J.L.; Muruganandan, S.; Dranse, H.J.; McMullen, N.M.; Sinal, C.J. Gpr1 is an active chemerin receptor influencing glucose homeostasis in obese mice. J. Endocrinol. 2014, 222, 201-215.

72.Scheja, L.; Heeren, J. The endocrine function of adipose tissues in health and cardiometabolic disease. Nat. Rev. Endocrinol. 2019, 15, 507-524.

83.Faienza, M.F.; Luce, V.; Ventura, A.; Colaianni, G.; Colucci, S.; Cavallo, L.; Grano, M.; Brunetti, G. Skeleton and glucose metabolism: a bone-pancreas loop. Int. J. Endocrinol. 2015, 758148.

84.Srichomkwun, P.; Houngngam, N.; Pasatrat, S.; Tharavanij, T.; Wattanachanya, L.; Khovidhunkit, W. Undercarboxylated osteocalcin is associated with insulin resistance, but not adiponectin, during pregnancy. Endocrine 2016, 53, 129–135.

96.Yoon, M.J.; Yoshida, M.; Johnson, S.; Takikawa, A.; Usui, I.; Tobe, K.; Nakagawa, T.; Yoshino, J.; Imai, S. SIRT1-Mediated eNAMPT Secretion from Adipose Tissue Regulates Hypothalamic NAD+ and Function in Mice. Cell Metab. 2015, 21, 706-717.

97.Wnuk, A.; Stangret, A.; Wątroba, M.; Płatek, A.E.; Skoda, M.; Cendrowski, K.; Sawicki, W.; Szukiewicz, D. Can adipokine visfatin be a novel marker of pregnancy‐related disorders in women with obesity? Obes. Rev. 2020, 21.

100.Lobo, T.F.; Torloni, M.R.; Mattar, R.; Nakamura, M.U.; Alexandre, S.M.; Daher, S. Adipokine levels in overweight women with early-onset gestational diabetes mellitus. J. Endocrinol. Invest. 2019, 42, 149-156.

111.Hida, K.; Wada, J.; Eguchi, J.; Zhang, H.; Baba, M.; Seida, A.; Hashimoto, I.; Okada, T.; Yasuhara, A.; Nakatsuka, A.; Shikata, K.; Hourai, S.; Futami, J.; Watanabe, E.; Matsuki, Y.; Hiramatsu, R.; Akagi, S.; Makino, H.; Kanwar, Y.S. Visceral adipose tissue-derived serine protease inhibitor: a unique insulin-sensitizing adipocytokine in obesity. Proc. Natl. Acad. Sci. USA 2005, 102, 10610-10615.

120.Boström, P.; Wu, J.; Jedrychowski, M.P.; Korde, A.; Ye, L.; Lo, J.C.; Rasbach, K.A.; Boström, E.A.; Choi, J.H.; Long, J.Z.; Kajimura, S.; Zingaretti, M.C.; Vind, B.F.; Tu, H.; Cinti, S, Højlund, K, Gygi, SP, Spiegelman, B.M. A PGC1-α-dependent myokine that drives brown-fat-like development of white fat and thermogenesis. Nature 2012, 481, 463-468.

144.Telejko, B.; Kuzmicki, M.; Wawrusiewicz-Kurylonek, N.; Szamatowicz, J.; Nikolajuk, A.; Zonenberg, A.; Zwierz-Gugala, D.; Jelski, W.; Laudański, P.; Wilczynski, J.; Kretowski, A.; Gorska, M. Plasma apelin levels and apelin/APJ mRNA expression in patients with gestational diabetes mellitus. Diabetes Res. Clin. Pract. 2010, 87, 176-183.

149.Furuhashi, M.; Saitoh, S.; Shimamoto, K.; Miura, T. Fatty Acid-Binding Protein 4 (FABP4): Pathophysiological Insights and Potent Clinical Biomarker of Metabolic and Cardiovascular Diseases. Clin. Med. Insights Cardiol. 2015, 8, 23-33.

151.Rodríguez-Calvo, R.; Girona, J.; Alegret, J.M.; Bosquet, A.; Ibarretxe, D.; Masana, L. Role of the fatty acid-binding protein 4 in heart failure and cardiovascular disease. J. Endocrinol. 2017, 233.

164.Li, L.; Lee, S.J.; Kook, S.Y.; Ahn, T.G.; Lee, J.Y.; Hwang, J.Y. Serum from pregnant women with gestational diabetes mellitus increases the expression of FABP4 mRNA in primary subcutaneous human pre-adipocytes. Obstet. Gynecol. Sci. 2017, 60.

166.Reinehr, T.; Woelfle, J.; Wunsch, R.; Roth, C.L. Fibroblast growth factor 21 (FGF-21) and its relation to obesity, metabolic syndrome, and nonalcoholic fatty liver in children: a longitudinal analysis. J. Clin. Endocrinol. Metab. 2012, 97, 2143-2150.

167.Dekker Nitert, M.; Barrett, H.L.; Kubala, M.H.; Scholz Romero, K.; Denny, K.J.; Woodruff, T.M.; McIntyre, H.D.; Callaway, L.K. I. Increased placental expression of fibroblast growth factor 21 in gestational diabetes mellitus. J. Clin. Endocrinol. Metab. 2014, 99, 2014.

169.Wang, D.; Zhu, W.; Li, J.; An, C.; Wang, Z. Serum Concentrations of Fibroblast Growth Factors 19 and 21 in Women with Gestational Diabetes Mellitus: Association with Insulin Resistance, Adiponectin, and Polycystic Ovary Syndrome History. PLoS ONE 2013, 8, e81190.

182.Scotece, M.; Conde, J.; Abella, V.; Lopez, V.; Lago, F.; Pino, J.; Gómez-Reino, J.J.; Gualillo, O. NUCB2/nesfatin-1: a new adipokine expressed in human and murine chondrocytes with pro-inflammatory properties, an in vitro study. J. Orthop. Res. 32,653–660.

183.Fasshauer, M.; Bluher, M.; Stumvoll, M. Adipokines in gestational diabetes. Lancet Diabetes Endocrinol. 2014, 2, 488–499.

184.Iliodromiti, S.; Sassarini, J.; Kelsey, T.W.; Lindsay, R.S.; Sattar, N.; Nelson, S.M. Accuracy of circulating adiponectin for predicting gestational diabetes: a systematic review and meta-analysis. Diabetologia 2016, 59, 692–699.

186.Wang, Y.; Lam, K.S.; Kraegen, E.W.; Sweeney, G.; Zhang, J.; Tso, A.W.; Chow, W.S.; Wat, N.M.; Xu, J.Y.; Hoo, R.L.; Xu, A. Lipocalin-2 is an inflammatory marker closely associated with obesity, insulin resistance, and hyperglycemia in humans. Clin. Chem. 2007, 53, 34–41.

We would like to take this opportunity to thank you for all the valuable and highly perceptive remarks which have definitely made a substantial contribution to the quality of our paper.

Yours faithfully,

Dr. Monika Ruszała and Assoc. Prof. Zaneta Kimber-Trojnar

Chair and Department of Obstetrics and Perinatology

Medical University of Lublin, 20-090 Lublin, Poland

Tel: +48-81-7244-769

E-mail: monika.ruszala@wp.pl

            zkimber@poczta.onet.pl

Reviewer 2 Report

This study examines the effect of 13 less known biomolecules in the pathogenesis of Gestational diabetes mellitus (GDM). The authors first described the well-known molecules with a proven important role in the pathogenesis of GDM, and then focus on 13 less known biomolecules.

I find this to a worthwhile and very important study in the field and I appreciate the thoroughness of the overall presented review manuscript and the summary in table 1.  However, I have several questions:

Q1: Page 3, lines 103 and 104. The authors referred that “galectin-1 and -3 are significantly dysregulated in the placentas of GDM patients”, but do not state directionality (increased or decreased). However, in lines 127-130 the authors refer that the role of galectin-1 in GDM is incomplete, which makes the first statement in lines 103-104 contradicting. Perhaps rephrasing the first sentence, or showing directionality (increased or decreased) instead of causality (dysregulated) would be beneficial.

Q2: Page 5, line 212. What do the authors mean with physiological pregnancy? Do the authors mean physiological changes during pregnancy? Or are they referring to normal/healthy pregnancy?

Q3: I think it would be helpful for the authors to describe in a schematic diagram the hypothesis and pathways they could propose within the reach of this review manuscript. It is certain that for some of the biomolecules there is still some information and direction lacking, but for others, the authors can hypothesize. Perhaps in a diagram, the authors can draw the well-known molecules (ex leptin), and add these new hypothesis. Moreover, the authors did a great work in describing the biomolecules individually, they should also summarize, integrate them and discuss them overall  in the conclusions.

 As such, I found that the authors present compelling results, however this manuscript should still have major revisions.

Minor revisions:

1- Line 51: Please change “In consequence excessive” to “In consequence, excessive”

2- Line 67: Please correct “during pregnancyseem”.

3- Lines 119-120: Repetitive wording. Please change the sentences “It also expresses anti-inflammatory actions [39]. Galectin 2 may be associated with elevated fasting glucose and fasting insulin levels. It is also involved”.

4- Line 145: please include citation “especially in patients with GDM.”.

5- Lines 171-175: I would reconsider some reformatting and rewording of the sentences.

Author Response

Dear Reviewer 2,

            We would like to resubmit our manuscript entitled “Novel biomolecules in the pathogenesis of gestational diabetes mellitus”. We appreciate your valuable remarks and hope that the quality of our manuscript is going to meet your expectations now that we have made some suggested alternations.

This study examines the effect of 13 less known biomolecules in the pathogenesis of Gestational diabetes mellitus (GDM). The authors first described the well-known molecules with a proven important role in the pathogenesis of GDM, and then focus on 13 less known biomolecules.

I find this to a worthwhile and very important study in the field and I appreciate the thoroughness of the overall presented review manuscript and the summary in table 1.”

Thank you very much for finding the time to read our manuscript. Thank you for considering our manuscript. We find all your remarks spot on therefore we have made a point-by-point correction of the manuscript according to your suggestions.

“However, I have several questions:

Q1: Page 3, lines 103 and 104. The authors referred that “galectin-1 and -3 are significantly dysregulated in the placentas of GDM patients”, but do not state directionality (increased or decreased). However, in lines 127-130 the authors refer that the role of galectin-1 in GDM is incomplete, which makes the first statement in lines 103-104 contradicting. Perhaps rephrasing the first sentence, or showing directionality (increased or decreased) instead of causality (dysregulated) would be beneficial.”

Following your advice, we have rephrased:

“Some galectines such as galectin-1, and -3 were described to be significantly decreased in the placentas of GDM patients.”

The sentence “The role of galectin 1 in GDM is still incomplete.” was removed.

“Q2: Page 5, line 212. What do the authors mean with physiological pregnancy? Do the authors mean physiological changes during pregnancy? Or are they referring to normal/healthy pregnancy?”

We have rearranged this sentence as follows:

“A transient phenomenon of insulin resistance is observed during pregnancy in women who were healthy before conception.”

“Q3: I think it would be helpful for the authors to describe in a schematic diagram the hypothesis and pathways they could propose within the reach of this review manuscript. It is certain that for some of the biomolecules there is still some information and direction lacking, but for others, the authors can hypothesize. Perhaps in a diagram, the authors can draw the well-known molecules (ex leptin), and add these new hypothesis.”

Following your advice and in response to Question 3, we have introduced Figure 1.

“Hypotheses and pathways of these 13 less known biomolecules in the pathogenesis of GDM were presented in Figure 1.

Figure 1. Hypotheses and pathways of novel biomolecules in the pathogenesis of GDM. FABP4, Fatty Acid-Binding Protein 4; FGF21, Fibroblast Growth Factor 21; GDF-15, Growth Differentiation Factor 15; GDM, gestational diabetes mellitus; IL-6, interleukin 6; RBP4, Retinol Binding Protein 4; TNF-α, tumour necrosis factor-alpha”

“Moreover, the authors did a great work in describing the biomolecules individually, they should also summarize, integrate them and discuss them overall  in the conclusions.”

Following your advice, we have included these sentences:

“Taking into account the presented studies, it seems that

  • high levels of FABP4,
  • high levels of one form of osteocalcin (i.e. ucOC),
  • low levels of irisin

in the serum of pregnant women can be used as predictive markers in the diagnosis of GDM.

[…]

Currently, these molecules have no application in the treatment of GDM, which is usually based on a diet. If the diet is not sufficient, insulin or oral antidiabetic medicaments are used depending on specific national guidelines regarding management of GDM.

We have also decided to add this sentence into the study abstract: “It seems that high levels of FABP4, low levels of irisin, and high levels of under-carboxylated osteocalcin in the serum of pregnant women can be used as predictive markers in the diagnosis of GDM”.

“As such, I found that the authors present compelling results, however this manuscript should still have major revisions.

Minor revisions:

1- Line 51: Please change “In consequence excessive” to “In consequence, excessive”

We have corrected.

“2- Line 67: Please correct “during pregnancyseem”.

We have corrected.

3- Lines 119-120: Repetitive wording. Please change the sentences “It also expresses anti-inflammatory actions [39]. Galectin 2 may be associated with elevated fasting glucose and fasting insulin levels. It is also involved.”

We have modified this sentence as follows:

“The main role of galectin 2 is immunomodulation. It also expresses via anti-inflammatory actions [39]. Galectin 2 may be associated with elevated fasting glucose and fasting insulin levels. Furthermore, this biomolecule is also involved in direct insulin secretion of pancreatic beta-cells. Its concentration increases with elevated fasting glucose and fasting insulin levels.”

“4- Line 145: please include citation “especially in patients with GDM.”

We have included a citation “especially in patients with GDM [47].”

  1. Tang, M.; Luo, M.; Lu, W.; Wang, S.; Zhang, R.; Liang, W.; Gu, J.; Yu, X.; Zhang, X.; Hu, C. Serum growth differentiation factor 15 is associated with glucose metabolism in the third trimester in Chinese pregnant women. Diabetes Res. Clin. Pract. 2019, 156, 107823.

“5- Lines 171-175: I would reconsider some reformatting and rewording of the sentences.”

We have rearranged this sentence as follows:

“Initially, it is synthesized as an inactive preproprotein containing 163 amino acids. Thereafter, a hydrophobic messenger peptide, consisting of 20 amino acids, is detached, while the produced proprotein is released from the cell.”

We would like to take this opportunity to thank you for all the valuable and highly perceptive remarks which have definitely made a substantial contribution to the quality of our paper.

Yours faithfully,

Dr. Monika Ruszała and Assoc. Prof. Zaneta Kimber-Trojnar

Chair and Department of Obstetrics and Perinatology

Medical University of Lublin, 20-090 Lublin, Poland

Tel: +48-81-7244-769

E-mail: monika.ruszala@wp.pl

            zkimber@poczta.onet.pl

Round 2

Reviewer 2 Report

The authors addressed the questions.